# The complete 3-year dataset of 4STAR sky-scans from ORACLES 2016-2018

Logan T. Mitchell[1], Connor J. Flynn[1], Kristina Pistone[2,3], Samuel E. LeBlanc[2,3], K. Sebastian Schmidt[4,5], Jens Redemann[1]

[1]School of Meteorology, University of Oklahoma, Norman, 73072, USA
[2]Bay Area Environmental Research Institute, Moffett Field, 94035, USA
[3]NASA Ames Research Center, Moffett Field, 94035, USA
[4]Laboratory for Atmospheric and Space Physics, Boulder 80303, USA
[5]University of Colorado, Boulder, 80303, USA

*Correspondence to*: Logan T. Mitchell (log.mitch@ou.edu)

**Abstract.** The NASA ORACLES (ObseRvations of Aerosols above CLouds and their intEractionS) airborne field campaigns deployed a 4STAR (Spectrometer for Sky-Scanning, Sun-Tracking Atmospheric Research) instrument onboard a P-3 aircraft to measure columnar optical properties of biomass burning aerosol smoke plumes over the Southeast Atlantic Ocean from 2016 to 2018. Although 4STAR's retrievals of aerosol optical properties from direct solar irradiances and diffuse sky radiances were performed, analyzed, and compared against other field campaigns via Single Scattering Albedo (SSA) campaign medians by Pistone et al., 2019 for ORACLES 2016, such an analysis was not extended to 2017 and 2018 due to previously unquantified instrument performance issues. As a result, only the 4STAR 2016 dataset was available to the public via https://doi.org/10.5067/Suborbital/ORACLES/P3/2016_V3 (ORACLES Science Team, 2021a). The instrument issues were diagnosed and mitigated through use of a four-wavelength set, instead of the previous five-wavelength set. Uniform Quality Control (QC) standards were established to ensure consistent data quality across all three campaigns. This resulted in research-quality, four-wavelength 4STAR datasets for 2017 and 2018 that have since been archived along with the original five-wavelength 4STAR 2016 dataset on the NASA Earth Science Project Office website, replacing the older versions at https://doi.org/10.5067/Suborbital/ORACLES/P3/2017_V3 (ORACLES Science Team, 2021b) and https://doi.org/10.5067/Suborbital/ORACLES/P3/2018_V3 (ORACLES Science Team, 2021c). The four-wavelength 4STAR 2016 dataset, although not on the archival site, is also publicly available via https://doi.org/10.5281/zenodo.16933793 (Mitchell, 2025). Potential improvements to these initial releases, such as broadening the spectral range, substituting for missing flight-level albedo, and removing unreliable scattering angles, are discussed. The complete 3-year ORACLES 4STAR 2016-2018 has many uses, including the determination of subseasonal changes in aerosol properties, modelling aerosol evolution, and the validation of satellite-retrieved aerosol products.

# 1 Introduction

## 1.1 ORACLES

ORACLES (ObseRvations of Aerosols above CLouds and their intEractionS) was a series of NASA airborne field campaigns to study Biomass Burning Aerosols (BBA) over the Southeast Atlantic Ocean (SEA) from 2016 to 2018. Each campaign was approximately a month long and occurred during September 2016, August 2017, and October 2018, collectively covering the peak of Southern Africa's BBA emission season (Redemann et al., 2021). BBA smoke plumes are transported from mainland Africa to the SEA by the Southern African Easterly Jet (Adebiyi and Zuidema, 2016), where the BBA interacts with a semi-permanent subtropical stratocumulus cloud deck (Sakaeda et al., 2011). These aerosol-cloud interactions are considered a major source of uncertainty for climate modelling of the region (Zuidema, 2016; Brown et al., 2021), in that models show a large spread in predictions and exhibit significant discrepancies relative to observations in this region.

## 1.2 4STAR

4STAR (Spectrometer for Sky-Scanning, Sun-Tracking Atmospheric Research) is a sun/sky spectrophotometer measuring direct solar irradiances and diffuse sky radiances (Dunagan et al., 2013). During ORACLES, it was mounted to the top of the NASA P-3 Orion aircraft. 4STAR observed above-cloud, below-plume columnar aerosol properties (Pistone et al., 2019; LeBlanc et al., 2020). 4STAR measures angularly-resolved sky radiances via two scanning geometries: principal plane (PPL) scans over a range of elevation angles with a fixed azimuth angle, and almucantar (ALM) scans consisting of azimuthal scans with a fixed elevation angle. Almucantar scans were conducted in pairs, with semicircular clockwise (CW) and counterclockwise (CCW) legs on either side of the sun (Dunagan et al., 2013). 4STAR sky radiance measurements were calibrated in the laboratory using an NIST (National Institute of Standards and Technology) referenceable 12-lamp 36-inch integrating sphere (Brown et al., 2005). 4STAR direct beam measurements were calibrated via refined Langley regressions (Schmid and Wehrli, 1995) conducted at Mauna Loa Observatory, whereby ground-based solar irradiance measurements are normalized by top-of-atmosphere irradiance during clear-sky conditions, removing Rayleigh extinction and background atmospheric effects. Calibrations generally bracket the campaigns, with sky radiance calibrations in March 2016, November 2016, June 2017, and November 2017; and direct beam calibrations in June 2016, November 2016, May 2017, February 2018, and August 2018. The 4STAR instrument was designed as a complement to AERONET (AErosol RObotic NETwork), which is a confederated network of sun/sky photometers for the observation of aerosol columnar properties. Measurement principle and deployment on an airborne platform allow 4STAR to serve as a mobile AERONET station.

We utilized an aerosol inversion code adapted from AERONET version 2.0 (Holben et al., 2006; Dubovik and King, 2000; Holben et al., 1998) to retrieve aerosol properties. In addition to the direct solar irradiances and diffuse sky radiances measured by 4STAR, the retrieval also requires flight-level albedo, which is calculated from SSFR (Solar Spectral Flux Radiometer) measurements of nadir upwelling and zenith downwelling spectral irradiances (Coddington et al., 2008).

The code has three stages: preparation of the inversion input, running the microphysical inversion, and extension of the retrieved microphysics to optical properties. In the first stage, Aerosol Optical Depth (AOD) spectral fitting and correction (subtraction of Rayleigh scattering and absorbing gases) occurs (Fig. 1a-b) and sky radiances are measured as a function of scattering angle (Fig. 1c). Additional first stage processing includes calculating SSFR flight-level albedo, plotting flight telemetry data, adjusting CW/CCW legs for ALM scans, and adjusting scattering angles/elevation angles for PPL scans. In the second stage, phase functions are determined via a damped least squares method, whereby sky radiances are fit iteratively as a function of scattering angle (Fig. 2a-b). This allows for calculations of the retrieval's primary outputs, the aerosol size distribution (Fig. 2c) and spectral complex refractive indices (Fig. 2d), until a final retrieval minimizing error is found. From there, aerosol radiative properties (the code's secondary outputs) are calculated (Fig. 3), including Single Scattering Albedo (SSA), AOD, Aerosol Absorption Optical Depth (AAOD), Extinction Ångström Exponent (EAE), Scattering Ångström Exponent (SAE), and Absorption Ångström Exponent (AAE). 4STAR sky-scans are compiled into daily NetCDF-3 files and archived on the NASA Earth Science Project Office (ESPO) website. The newest versions of these datasets (presented in the Methods section) are also available on Zenodo (Mitchell, 2025), both in daily NetCDF-3 files, as well as individual NetCDF-3 files of each sky-scan. 4STAR's spatiotemporal coverage during the ORACLES campaigns and the variable naming convention used within the files can be found in this paper's supplement.

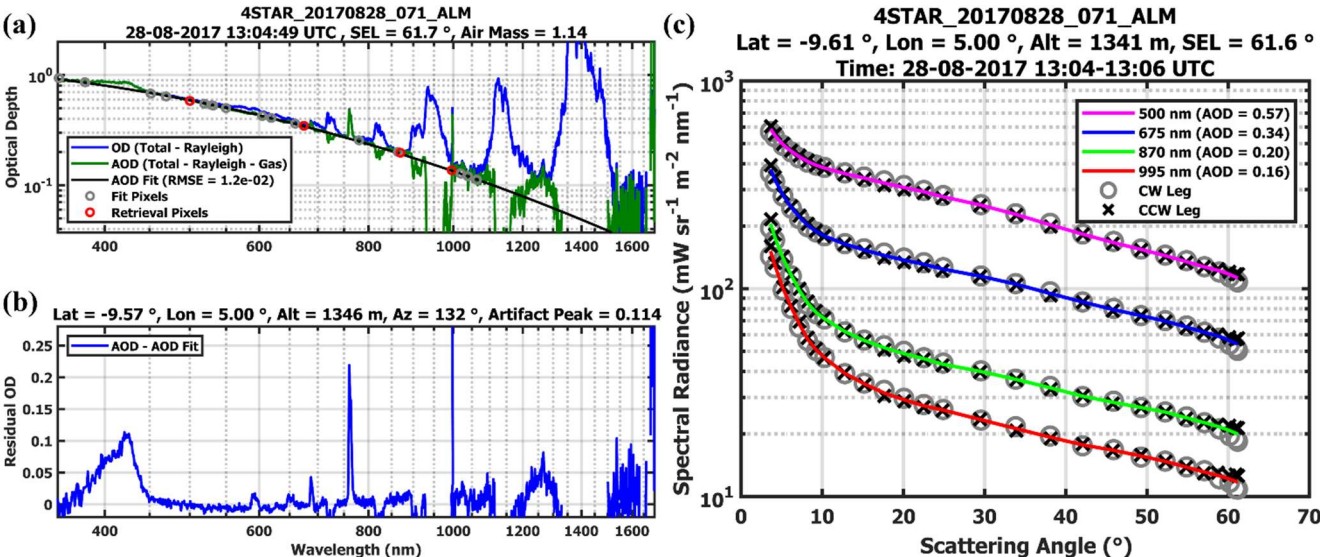

**Figure 1: (a) AOD fitting, (b) AOD residuals, and (c) sky radiance averaging for a four-wavelength ALM sky-scan from ORACLES 2017. These figures are generated during the retrieval's first stage. In (a), optical depth (OD) is calculated by subtracting Rayleigh scattering from total optical depth. AOD is calculated by subtracting Rayleigh scattering and gas absorption ($CH_4$, $CO_2$, $NO_2$, $O_2$-$O_2$, and $O_3$) from total optical depth. The AOD fit is a quadratic polynomial of AOD over wavelength in log-log space for 36 pixel wavelengths. The (b) difference between AOD and the AOD fit shows an instrument artifact peak of 0.114 at 420 nm and the absorbing gas bands. Despite the presence of the instrument artifact, the four-wavelength selection allowed for a valid retrieval of aerosol properties (see Fig. 2). The (c) sky radiances at four wavelengths show strong agreement between the CW and CCW legs, confirming the uniformity of sky conditions. Listed metadata include Latitude (Lat), Longitude (Lon), Altitude (Alt), Azimuthal Angle (Az), and Solar Elevation Angle (SEL).**

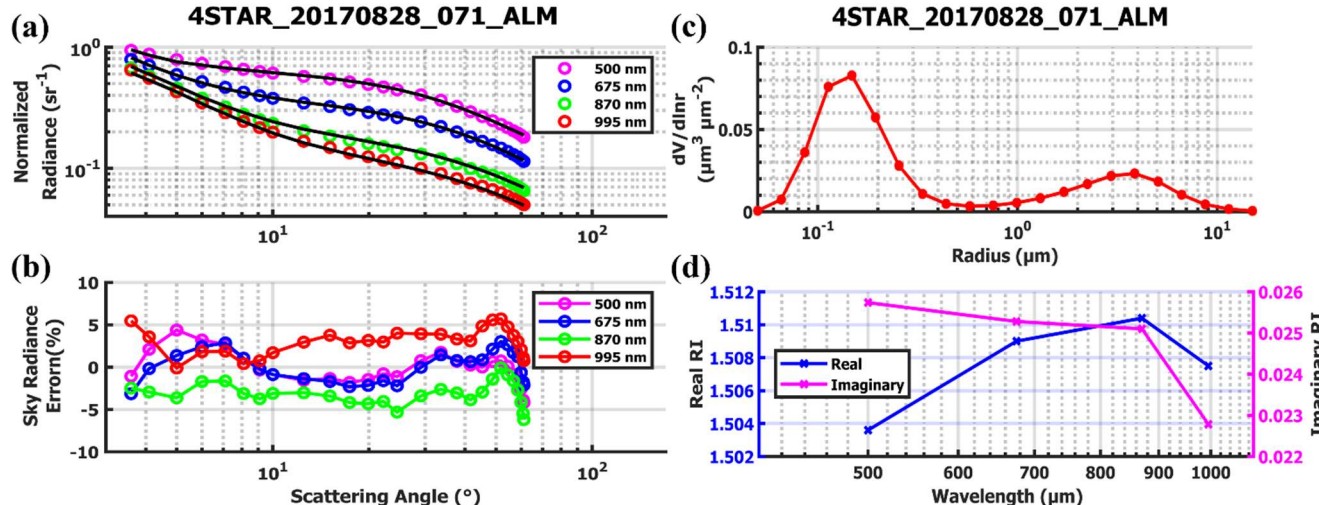

Figure 2: (a) Sky radiance fitting, (b) sky radiance error, (c) size distribution, and (d) real and imaginary refractive indices for a four-wavelength ALM sky-scan from ORACLES 2017. These figures are generated during the retrieval's second stage. The (a) sky radiances are fit across scattering angles, with the (b) difference between the measurement and fit within ±7 % for this sky-scan. The (c) size distribution is dominated by the fine mode, which is indicative of the BBA being studied. The (d) real Refractive Index (RI) represents aerosol refraction, while the imaginary RI represents aerosol attenuation.

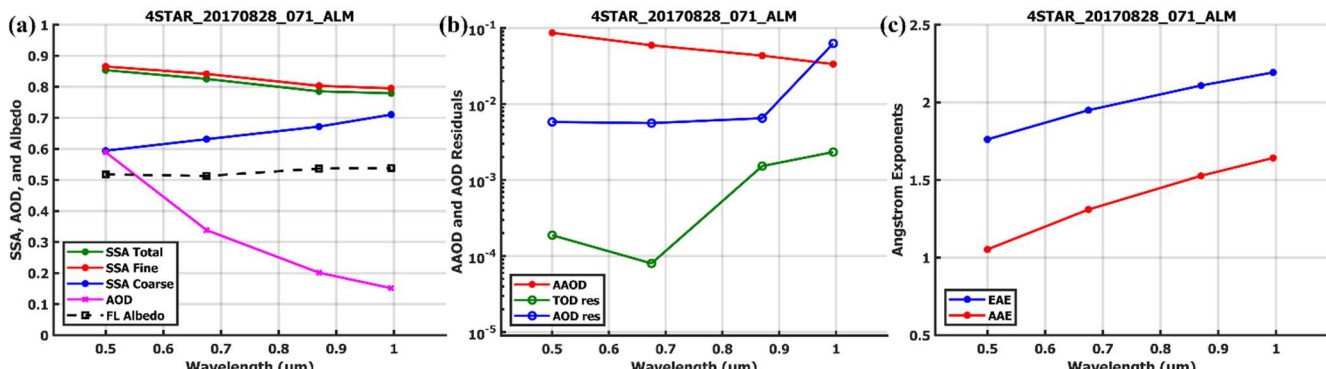

Figure 3: (a) SSA and AOD, (b) AAOD and AOD residuals, and (c) EAE and AAE for a four-wavelength ALM sky-scan from ORACLES 2017. These figures are generated during the retrieval's third stage. SSA (a) is expressed in terms of the total, as well as its fine and coarse components, with the fine mode again being dominant. In addition, retrieved AOD and SSFR-derived flight-level (FL) albedo are also plotted. AAOD (b) is the absorption component of AOD, while Total Optical Depth (TOD) residual (res) is calculated by subtracting retrieved TOD from input TOD, and AOD res is the AOD fit subtracted from the AOD measurement.

### 1.2.1 Five-Wavelength Selection

Pistone et al. (2019) developed a system for the processing, retrieval, and screening of 4STAR sky-scans for the ORACLES 2016 campaign. A five-wavelength (5wl) set of 400, 500, 675, 870, 995 nm was selected to align 4STAR with AERONET inversion wavelengths (440, 675, 870, 1020 nm) as closely as possible within the constraints of 4STAR spectrometer operation. The longest wavelength provided by the 4STAR UV/VIS (Ultraviolet/Visible) spectrometer is 995

nm, while wavelengths of 400 and 500 nm were selected to avoid an instrument sensitivity near 440 nm that caused anomalously low SSA values.

### 1.2.2 Manual QC Criteria

To screen the 4STAR sky-scans from ORACLES 2016, Pistone et al. (2019) adapted Quality Control (QC) criteria from AERONET, described in Table 1. The QC criteria presented here ensure: (1) sufficient aerosol loading ($AOD_{400} > 0.2$), (2) level flight altitude, (3) low error between measured sky radiances and the inversion results, (4) sufficient scattering angles spanning the critical range, (5) manual inspection for uniform and cloud-free conditions during the sky scan, (6) a sufficiently low altitude to capture a full vertical view of the smoke plume, and (7) an additional flag to identify high aerosol

loadings ($AOD_{400} > 0.4$). Passing criteria (1-5) was sufficient for archival on the NASA ESPO website (ORACLES Science Team, 2021a), while criterion (6) was used in the Pistone et al. (2019) analyses that compared these retrievals with other full-column instruments, and criterion (7) was included to best approximate AERONET level 2 requirements (although still including PPL scans).

**Table 1: Four recommended selections of 4STAR sky-scans for common research scenarios, following the application of the Pistone et al. (2019) manual QC criteria to the ORACLES 2016 campaign. R1 indicates that the five-wavelength 2016 dataset is its secondary release on the ESPO site. Four sky-scans erroneously included in R0 were removed and QC flags for criteria 6 and 7 were added, but the dataset is otherwise unchanged.**

| Manual QC Criteria | | | Selection #1 Loading: Moderate Altitude: Any Purpose: All high-quality sky-scans archived on ESPO. | Selection #2 Loading: Moderate Altitude: Low Purpose: Full vertical view of the smoke plume. | Selection #3 Loading: High Altitude: Any Purpose: Proxy for AERONET standards. | Selection # 4 Loading: High Altitude: Low Purpose: Full vertical view and AERONET proxy. |
|---|---|---|---|---|---|---|
| 1. AOD (400 nm) > 0.2 <br> 2. Altitude difference < 50 m <br> 3. Mean sky radiance error (\|meas - fit\|) < 10 % <br> 4. Scattering angles minimally span 3-50 ° <br> 5. Inspection of sky error per scattering angle <br> 6. Altitude < 3 km <br> 7. AOD (400 nm) > 0.4 | | | (green bar) | (blue bar) | (red bars) | (purple bar) |
| *Name* | *Total Sky-Scans* | *Converged Retrievals* | *Selection #1* | *Selection #2* | *Selection #3* | *Selection #4* |
| R1_2016_5wl | N = 174 | 164 (94 %) | 82 (47 %) | 75 (43 %) | 72 (41 %) | 68 (39 %) |

### 1.2.3 Research Goals

The goal of this study is to describe the newly-produced 4STAR retrievals for ORACLES 2017 and 2018, replacing older versions on the ESPO site (ORACLES Science Team, 2021b; ORACLES Science Team, 2021c). Since ORACLES 4STAR 2016 data has been used extensively in the peer reviewed literature (Cochrane et al., 2019; LeBlanc et al., 2020; Cochrane et al., 2021, Pistone et al., 2021; Cochrane et al., 2022), we expect this extension of 4STAR sky-scans through 2018 to be a welcome contribution to the ORACLES dataset. Because of a 4STAR instrument artifact and stray light

scattering that occurred starting in 2017, it is not sufficient to simply apply the same wavelength selection and manual QC

criteria that Pistone et al. (2019) used for ORACLES 2016. Instead, a new wavelength set must be selected that avoids the above two issues, and automated QC criteria developed to handle the larger sky-scan totals for ORACLES 2017 and 2018. By applying the same standards to all three campaigns, another dataset for ORACLES 2016 is created (Mitchell, 2025), resulting in a comprehensive, uniform inversion product for the entire campaign. Although this dataset will not replace the current iteration on the ESPO website, it is still useful for comparing the two methodologies, and its uniformity with ORACLES 2017 and 2018 makes it ideal for the subseasonal analyses conducted by the authors in a separate study. In addition, we expect the 4STAR 2017 and 2018 datasets to be used as vigorously as the 4STAR 2016 dataset in satellite and climate model validation studies.

## 2 Methods

### 2.1 Four-Wavelength Selection

The first step toward extending 4STAR retrievals to the ORACLES 2017 and 2018 campaigns was wavelength selection. An instrument artifact near 420 nm was discovered in 2017 that persisted in 2018, causing anomalously high AOD and affecting 400 nm measurements. To avoid this issue, while simultaneously extending 4STAR's retrievals into the ultraviolet spectrum, replacing 400 nm with 360 or 380 nm was explored. If successful, these ultraviolet retrievals would be useful for the identification of Brown Carbon (BrC) versus Black Carbon (BC) in the smoke plume (Russell et al., 2010). BrC has an AAOD with a strong spectral dependence (AAE > 1.5) in the near ultraviolet range, as contrasted with BC's uniform AAOD spectral dependence (AAE ~ 1). The optical identification of chemically-similar BrC and BC could prove critical for improving aerosol parameterization in climate models.

While it appeared that 360 and 380 nm were not affected by the instrument artifact near 420 nm, the AOD uncertainty was much higher at these wavelengths than for those selected by Pistone et al. (2019), thus precluding their use in these retrievals. We attribute this uncertainty to stray light scattering (Zong et al., 2006), which appears to be internal to the spectrometer and primarily impactful in the UV spectrum due to the relatively low intensities of short wavelength light. The stray light scattering issue is apparently intermittent, while the instrument artifact's effect on AOD varies in both magnitude and direction. As such, a four-wavelength (4wl) set of 500, 675, 870, and 995 nm was selected to avoid both issues. These four wavelengths were again selected to best align 4STAR with AERONET inversion wavelengths (440, 675, 870, 1020 nm), limited to the longest wavelength provided by the 4STAR UV/VIS (Ultraviolet/Visible) spectrometer of 995 nm, and avoiding the instrument artifact near 420nm by using 500 nm. The effect of removing 400 nm was examined by comparing AOD and SSA from the new four-wavelength set to that of the original five-wavelength set. A potential method for expanding 4STAR retrievals into the UV spectrum is explored in the discussion section.

## 2.2 Automated QC Criteria

The next step was developing an automated QC criteria (Table 2) to handle the larger sky-scan totals of ORACLES 2017 and 2018, while remaining compatible with the manual criteria for ORACLES 2016 from Pistone et al. (2019). (1-3) are the same, while (4-7) similarly ensure consistent measurements at critical scattering angles and check for data gaps. The covariance matrix sky error is utilized for (8) as this value allows us to identify large differences between measured sky radiances and the fit at any scattering angle. This indicates if there are uniform aerosol conditions, removing the need for manual inspection of sky error as a function of scattering angle. The two new criteria ensure: (9) stable flight telemetry and (10) the retrieval is not on the boundary limits of the parameter space. In the same vein as the final two criteria from Table 1, (11) is again for low-altitude scans with a full vertical view of the smoke plume and (12) for high aerosol loadings. (1-10) are required for archival on the ESPO site, while (11) and (12) are optional criteria that we developed for specific research goals. The QC status of every 4STAR sky-scan per ORACLES campaign dataset can be found in the supplement.

**Table 2: Four recommended selections of 4STAR sky-scans for common research scenarios, following the application of the new automated QC criteria to the ORACLES 2016-2018 campaigns. R0 indicates that the four-wavelength 2017 and 2018 datasets are their initial releases on the ESPO site. T0 represents "initial testing", as the four-wavelength 2016 dataset will not be archived on the ESPO site. An * implies that this value is only over the critical scattering angle range of 3.5 - 30 °.**

| Automated QC Criteria | Selection #1 Loading: Moderate Altitude: Any Purpose: All high-quality sky-scans archived on ESPO. | Selection #2 Loading: Moderate Altitude: Low Purpose: Full vertical view of the smoke plume. | Selection #3 Loading: High Altitude: Any Purpose: Proxy for AERONET standards. | Selection #4 Loading: High Altitude: Low Purpose: Full vertical view and AERONET proxy. |
|---|---|---|---|---|
| 1. AOD (400 nm) > 0.2<br>2. Altitude difference < 50 m<br>3. Mean sky radiance error (\|meas - fit\|) < 10 %<br>4. Minimum scattering angle < 6 °<br>5. Maximum scattering angle > 50 °<br>6. Mean scattering angle difference < 3 ° *<br>7. Maximum scattering angle difference < 10 ° *<br>8. Covariance matrix sky error < 10 %<br>9. Roll standard deviation < 3 °<br>10. Passes retrieval boundary test<br>11. Maximum altitude < 3 km<br>12. AOD (400 nm) > 0.4 | | | | |

| Name | Total Sky-Scans | Converged Retrievals | Selection #1 | Selection #2 | Selection #3 | Selection #4 |
|---|---|---|---|---|---|---|
| T0_2016_4wl | N = 174 | 163 (94 %) | 88 (51 %) | 77 (44 %) | 72 (41 %) | 68 (39 %) |
| R0_2017_4wl | N = 381 | 346 (91 %) | 157 (41 %) | 141 (37 %) | 92 (24 %) | 85 (22 %) |
| R0_2018_4wl | N = 230 | 224 (97 %) | 96 (42 %) | 92 (40 %) | 49 (21 %) | 49 (21 %) |

## 3 Results

### 3.1 Wavelength Selection

We examined the effect of changing from a five-wavelength to a four-wavelength retrieval set by observing the resultant differences in AOD and SSA for ORACLES 2016 at the four overlapping wavelengths (Fig. 4). The 75 sky-scans

from Pistone et al. (2019) were used, although the change to the four-wavelength set caused one sky-scan to fail to converge on a retrieval, resulting in only 74 sky-scans for the comparison. Across all four overlapping wavelengths, 97 % of AOD differences are within ±0.001, while 100 % fall within ±0.005. This shows that the differences between the five-wavelength and four-wavelength sets are well within the 4STAR AOD uncertainty of ±0.01 from Pistone et al. (2019) and LeBlanc et al. (2020). For SSA, the median difference at 500 nm is -0.0045, indicating that the new wavelength selection generally results

in a slight decrease in SSA at that wavelength. At the other wavelengths, the median differences are centered near zero with tighter distributions. Thus, removing 400 nm marginally affects the retrieval of aerosol radiative properties at 500 nm, with negligible effects at longer wavelengths. 500 nm is most affected because it had previously been constrained by shorter wavelength values at 400 nm in the five-wavelength set but is now the shortest retrieved wavelength in the four-wavelength set. The small negative shift in SSA at 500 nm appears to be systematic, as the SSA difference for about 80 % of retrievals is

less than zero. Thus, one should expect an average bias of about -0.004 when comparing SSA at 500 nm from the four-wavelength set to the five-wavelength set.

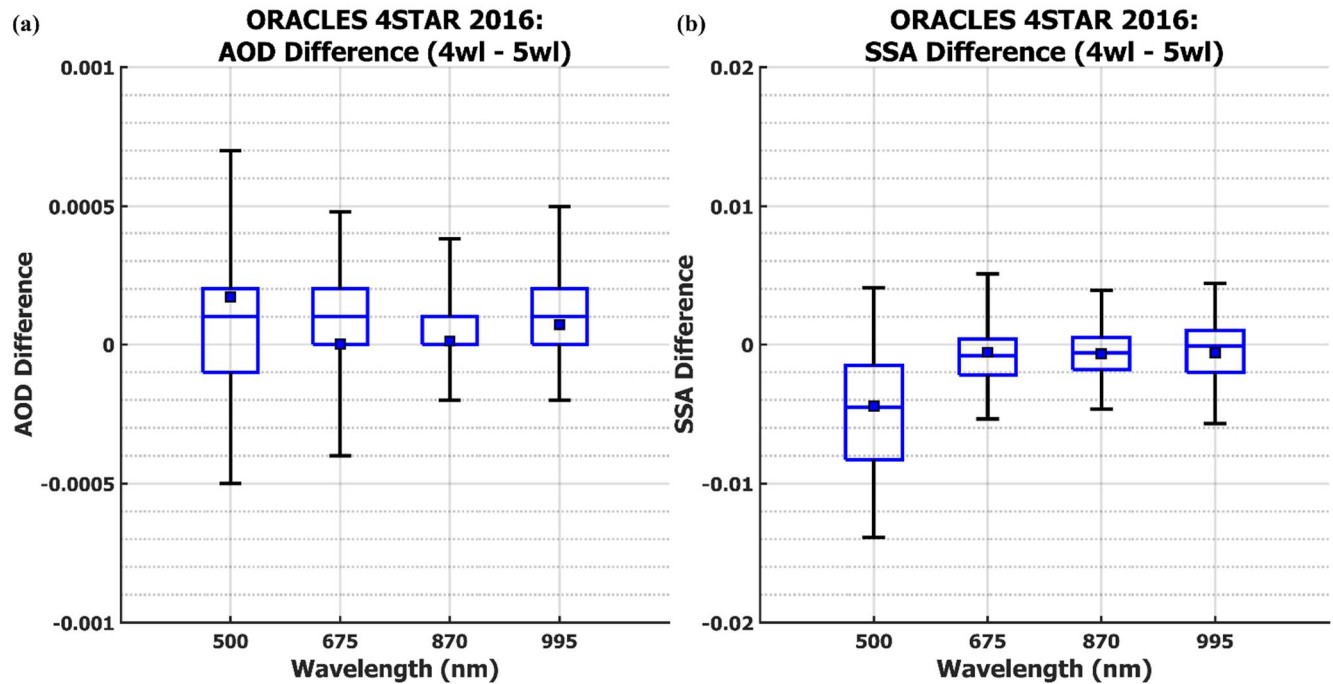

**Figure 4: (a) AOD and (b) SSA differences between the four-wavelength (4wl) and five-wavelength (5wl) datasets for ORACLES 4STAR 2016. Boxes show the interquartile range, while whiskers extend to the 5th and 95th percentiles. Central lines are medians**
**and squares are means.**

### 3.2 Automated QC Criteria

The application of the automated QC criteria creates research-quality 4STAR sky-scan datasets for all three years of ORACLES. For ORACLES 2017 and 2018, these datasets are their initial releases on the ESPO site. For ORACLES 2016,

the new four-wavelength dataset will not replace the original five-wavelength dataset but is useful for comparison with the data from ORACLES 2017 and 2018 and thus is also publicly available (Mitchell, 2025).

The use of optional criteria (11) and (12) allows for researchers to further refine the 4STAR datasets for their purposes. This results in four recommended Selections for researchers: (S1) all research-quality sky-scans, (S2) low-altitude sky-scans ensuring retrieval of the full-column smoke plume, (S3) high aerosol loading sky-scans to best align with AERONET standards, or (S4) sky-scans meeting all the above QC criteria.

For (S1) of ORACLES 2016, 75 sky-scans met both the Pistone et al. (2019) manual QC criteria and the automated QC criteria. Seven sky-scans meeting the manual criteria were excluded by the automated criteria – three lacking scattering angles < 6 °, three with covariance matrix sky error > 10 %, and one ALM scan that fails to converge on a retrieval. Conversely, thirteen sky-scans removed during manual inspection met all automated QC criteria.

Additionally, we note that there are no (S1) sky-scans for seven research flights of ORACLES 2018. Three research flights (21 October 2018, 23 October 2018, and 25 October 2018) only observed low aerosol loadings ($AOD_{400}$ < 0.2) over the Southeast Atlantic, preventing any sky-scans from meeting (S1) criteria. 4STAR retrievals were not generated at all for four research flights (24 September 2018, 27 September 2018, 2 October 2018, and 3 October 2018), due to the lack of SSFR measurements on those days. Possible solutions to compensate for the lack of SSFR flight-level albedo are discussed in the following section.

# 4 Discussion

## 4.1 Broader Context

4STAR-retrieved SSA campaign medians are compared to that of Pistone et al. (2019) and other previous studies in Fig. 5. This includes measurements from the AERONET station in Mongu, Zambia (15.254 °S, 23.151 °E) from Dubovik et al. (2002) and Eck et al. (2013). Southern African Regional Science Initiative (SAFARI-2000) was an airborne field campaign with a small number of flights taking place off the coast of Southern Africa. SAFARI-2000 employed the combination of Particle Soot Absorption Photometers (PSAP) with TSI nephelometers (neph) to measure in situ absorption and scattering coefficients, respectively (Haywood et al., 2003). This included sampling fresh smoke on 13 September 2000 and later resampling the same plume after it had aged for 2-3 days. Another SAFARI-2000 study (Russell et al., 2010) examined retrievals from AATS-14 (Ames Airborne Tracking Sun photometer at 14 wavelengths), a predecessor to 4STAR, which is also reliant on SSFR inputs. In addition to 4STAR retrievals and in situ PSAP and neph, Pistone et al. (2019) also included retrievals from AirMSPI (Airborne Multi-angle SpectroPolarimeter Imager), an imaging polarimeter employed on the high-flying ER-2 aircraft.

The SSA campaign medians are generally bounded by the Haywood et al. (2003) results, with the fresh plume indicating more absorption and the aged plume more scattering. There is good agreement between the ORACLES 4STAR

2016 campaign medians from this study and Pistone et al. (2019), especially at 670 and 875 nm. ORACLES 4STAR SSA medians decrease slightly from August to September, then greatly increase by October. This marked increase in SSA over the BBA emission season is the subject of a subseasonal analysis conducted by the authors. The slight decrease in SSA in September not found in other studies (Eck et al., 2013) will also be addressed by that analysis.

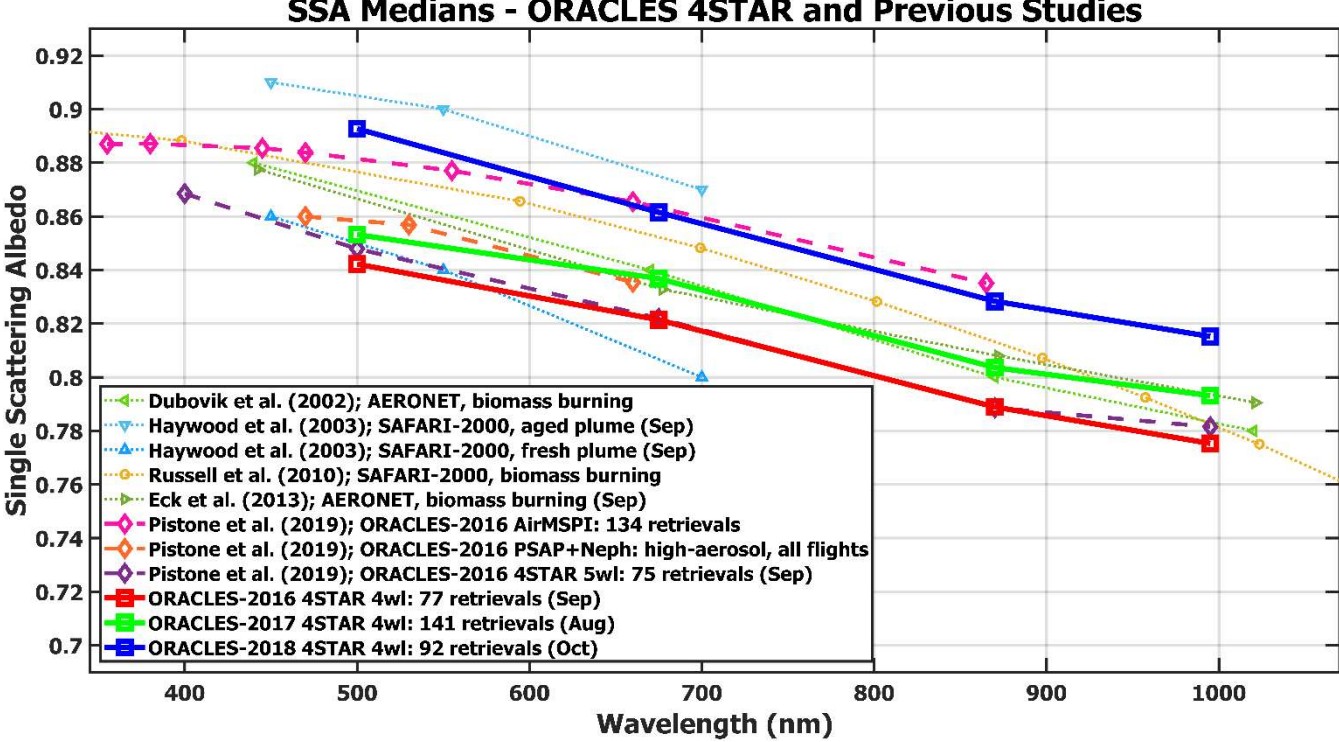

**Figure 5: SSA campaign medians from ORACLES 4STAR 2016-2018 and previous studies. Here, all 4STAR data meets S2 requirements. Adapted from Fig. 11b of Pistone et al. (2019).**

### 4.2 Practical Applications

In addition to comparison with previous campaigns, Pistone et al. (2019) also utilized 4STAR sky-scans to compare against aerosol properties derived from four other retrieval methods using ORACLES instrumentation: (1) SSFR spectral irradiance with 4STAR AOD in a radiative transfer model, (2) PSAP absorption with Neph scattering, (3) RSP (Research Scanning Polarimeter) total and polarized reflectance in a MAPP (Microphysical Aerosol Properties from Polarimetry) algorithm, and (4) AirMSPI polarized radiance in a radiative transfer model. Of these, the ORACLES 2016 SSA campaign medians for PSAP with Neph and AirMSPI are displayed in Fig. 5.

Pistone et al. (2019) conducted two case studies, on 12 September 2016 and 20 September 2016, to examine different available instrument comparisons. The first day allowed for the comparison of SSA, AOD, AAOD, EAE, SAE, and AAE from 4STAR, PSAP with Neph, and RSP; while the second day allowed for the comparison of the same properties from 4STAR, SSFR, PSAP with Neph, and AirMSPI. The authors found that SSA generally agreed within measurement

uncertainties across the different instruments. There was also stronger agreement between 4STAR and PSAP with Neph for higher total aerosol loading ($AOD_{400} > 0.4$). SSA, EAE, SAE, and AAE were all also compared for the entire ORACLES 2016 campaign between 4STAR, PSAP with Neph, and AirMSPI. The $SSA_{500}$ campaign medians from these three instruments all fell within the 0.85 to 0.88 range, in alignment with previous literature (see Fig. 5). The expansion of ORACLES 4STAR data to 2017 and 2018 will allow for an extension of case study comparisons with other instruments that also operated for all three ORACLES years, namely SSFR, PSAP with Neph, and RSP, as well as for campaign-wide comparisons with PSAP with Neph.

Jethva et al. (2024) made use of our new ORACLES 4STAR 2016-2018 dataset to validate aerosol properties from satellite retrievals. Above-Cloud Aerosol Optical Depth (ACAOD) from 4STAR and HSRL-2 was combined with Top-Of-Atmosphere (TOA) spectral reflectance from MODerate resolution Imaging Spectroradiometer (MODIS) and Ozone Monitoring Instrument (OMI) aboard the Terra/Aqua and Aura satellites, allowing for the retrieval of aerosol SSA and aerosol-corrected Cloud Optical Depth (COD). The retrieval assumes a color ratio effect, whereby TOA reflectance is lower at shorter wavelengths due to strong aerosol absorption over low clouds. The airborne and satellite retrievals of SSA generally agreed within 0.01, which is within inversion uncertainties. Thus, these retrievals of above-cloud aerosol properties are critical to the development of a global satellite-retrieved SSA product.

Fakoya et al. (2025) also utilized this dataset to model aerosol evolution over the Southeast Atlantic Ocean. Aerosol observations, including airborne 4STAR retrievals and ground-based AERONET retrievals, were combined with aerosol properties from WRF-CAM5 and aerosol age estimates from a regional model of WRF-AAM (WRF model coupled with Aerosol Aware Microphysics). Their results showed that fresh smoke aerosols near the fire emission source were initially smaller, highly absorbing particles. By excluding aerosols within the boundary layer, they found that free tropospheric aerosols generally decreased in absorptivity over the first six to seven days, but that absorptivity often increased after ten days, showing that aerosol absorption is not linearly affected by chemical aging processes. Such studies using the present data to better determine aerosol aging effects will thus help to reduce the uncertainty of future regional climate models of the Southeast Atlantic.

The entire ORACLES 4STAR 2016-2018 dataset is also the focus of a forthcoming publication by the authors, whereby the subseasonal dependence of aerosol properties, including SSA, AOD, AAOD, and real and imaginary refractive indices, are examined. The evolution of these aerosol properties over the Southeast Atlantic during Southern Africa's BBA emission season can show if aerosols are brightening or darkening and suggest changes in aerosol type or composition.

### 4.3 Potential Improvements

Although this study created complete 4STAR datasets for ORACLES 2016-2018, potential improvements can still be made. The most pressing issue is the treatment of the instrument artifact near 420 nm and stray light scattering in the near ultraviolet wavelengths. This can likely be achieved by running 4STAR retrievals through GRASP (Generalized Retrieval of Atmosphere and Surface Properties) code, which is developed at the University of Lille. The current AERONET-adapted

code is highly dependent upon wavelength selection, with the wavelength set affecting retrieved properties. Additionally, the retrieval code fails if selected wavelengths are too spectrally close, which can occur if they are within 10-20 nm of each other. However, the GRASP retrieval uses hundreds of wavelengths (Román et al., 2018), which is well-suited to the hyperspectral capabilities of 4STAR. With additional retrieval input from 350 to 500 nm, it will be easier to diagnose and remove the instrument artifact and stray light scattering issues via sensitivity testing. Retrieval of aerosol absorption in the UV wavelengths will also greatly assist in BrC identification. This was not implemented by the current study as to not further delay the release of research-quality 4STAR datasets for ORACLES 2017 and 2018 but should be explored in a future generation of these retrievals.

The lack of SSFR flight-level albedo for three research flights of ORACLES 2018 is currently limiting the number of successful 4STAR retrievals for that year. A possible solution involves the use of satellite-retrieved surface albedo as a replacement input on those days. This can include use of the 500 m global albedo product (MCD43A3) from MODIS aboard the Terra and Aqua satellites. Due to the product being limited to land and coastal waters, surface albedo must be extrapolated to flight coordinates, and corrections made for cloud albedo between the surface and flight-level.

A final potential improvement is rectifying high sky radiance error near the horizon. A code could be developed to examine sky radiance error as a function of scattering angle. If the sky radiance error is consistently greater than 10 % at higher scattering angles (above 80 °), then the scattering angles are removed and the retrieval re-run. This would result in an overall decrease in average sky radiance error and covariance matrix sky error, possibly allowing more sky-scans to meet QC criteria. Based on an inspection of sky-scans with high sky error above 80 ° that only fail due to sky error or retrieval boundary issues, we estimate that implementing this method could increase the number of sky-scans meeting QC criteria by about 3 sky-scans for 2016, 11 sky-scans for 2017, and 14 sky-scans for 2018. However, this method also has the potential to affect the sky-scans' retrieved aerosol properties, including SSA, in unquantified ways, making it inappropriate for an initial release.

**Data Availability:**

The NASA P-3 aircraft data are accessible via the NASA ESPO website:

ORACLES 2016: https://doi.org/10.5067/Suborbital/ORACLES/P3/2016_V3 (ORACLES Science Team, 2021a),

ORACLES 2017: https://doi.org/10.5067/Suborbital/ORACLES/P3/2017_V3 (ORACLES Science Team, 2021b),

ORACLES 2018: https://doi.org/10.5067/Suborbital/ORACLES/P3/2018_V3 (ORACLES Science Team, 2021c).

All four 4STAR datasets (R1_2016_5wl, R0_2017_4wl, R0_2018_4wl, and T0_2016_4wl) utilized in this paper, along with figure data, are available via https://doi.org/10.5281/zenodo.16933793 (Mitchell, 2025).

## 5 Conclusions

Here we have described a newly available product of aerosol retrievals making use of airborne data from three field deployments over the Southeast Atlantic Ocean. To accomplish this, we chose a new wavelength set and developed new automated QC criteria developed for the processing of 4STAR retrievals from ORACLES 2017 and 2018. The wavelength selection and QC criteria were designed to be as compatible with past work on the 2016 retrievals (Pistone et al. 2019) as possible, while avoiding the effects of a new instrument artifact and addressing greater sky-scan totals from the latter two years. This has resulted in the first-ever releases of four-wavelength 4STAR datasets for 2017 and 2018 that have joined the original five-wavelength 4STAR 2016 dataset on the ESPO archival site. The four-wavelength 4STAR 2016 dataset is also publicly available for research purposes and consistency with the 2017 and 2018 datasets. Based on the application of optional QC criteria, four recommended selections for researchers are presented. This includes all research-quality data, as well as subsets of low altitude sky-scans (full vertical view of the smoke plume), high aerosol loading sky-scans (approximating AERONET standards), and sky-scans that are both low altitude and high aerosol loading.

Our ORACLES 4STAR 2016 SSA campaign medians are within ±0.007 of the Pistone et al. (2019) set for the four overlapping wavelengths, putting them in alignment with other ORACLES measurements and previous studies in the SEA. Potential improvements can still be made, such as addressing stray light scattering and the instrument artifact via GRASP retrieval code, producing additional retrievals by adding satellite albedos to specific cases, and removing unreliable sky radiances near the horizon. This complete 3-year ORACLES 4STAR 2016-2018 dataset has numerous scientific applications, including determining subseasonal changes in BBA properties, modelling BBA evolution over the SEA, and validating satellite-retrieved aerosol products.

## Author Contributions

This dataset was created by LTM, under the guidance of CJF and JR, and with input from KP and SEL. JR and SEL were PIs for 4STAR during ORACLES 2016 and ORACLES 2017-2018, respectively. KSS was the PI for SSFR during ORACLES. KP and SEL operated the 4STAR instrument aboard the P-3 aircraft for all of ORACLES, while CJF also operated during ORACLES 2016. KP processed the ORACLES 2016 five-wavelength 4STAR data and created the manual QC criteria. LTM processed ORACLES 2017 and 2018 4STAR data, re-processed ORACLES 2016, and created the automated QC criteria. Figures 1-3 were created via code from CJF and edited by LTM. Tables 1-2 and Figure 4 were created by LTM. Figure 5 was adapted from a figure by KP to include the new datasets created by LTM. LTM prepared the manuscript with contributions from all co-authors.

## Competing Interests

The authors declare that they have no conflict of interest.

**Acknowledgements**

This research was supported by the NASA Atmosphere Observing System (AOS) mission (grant no. 80NSSC23M0083). It was also supported by the University of Oklahoma (OU) start-up package (grant no. 122007900). The ORACLES field campaign was funded through the NASA Earth Venture Suborbital-2 program (grant no. NNH13ZDA001N-EVS2). We

thank the NASA ORACLES team for a successful mission. Data analysis and visualization were conducted utilizing MATLAB.

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
