# Peer review of "The complete 3-year dataset of 4STAR sky-scans from ORACLES 2016-2018"

_Earth System Science Data, 2025_

## Author Comment (AC6)

RC1:

The manuscript is well-written and clearly expresses the methodology and results. I appreciated the authors efforts to provide a comprehensive product for the entire 3 years of the campaign, especially given the issues with an instrument artifact. Providing the data both on the ESPO website and in Zenodo seemed like a logical choice to increase the accessibility of the data. Researchers familiar with the data would go to the ESPO website while others may find it in Zenodo. It would be good to provide more context about the data on the Zenodo page. I would recommend linking to this manuscript in the Zenodo metadata so that researchers can learn more about the context of the data.

Response:

The preprint DOI has been linked on Zenodo.

RC2:

This manuscript provides the data from the NASA ORACLES airborne filed campains that deployed a 4STAR instrument onboard a P-3 aircarft to measure columnar optical properties of biomass burning aerosol smoke plumes over the Southeast Atlantic Ocean from 2016 to 2018. The authors well describe the wavelength selection and quality control criteria, however, I think it is better to show a period of the data or a single case for example may help the readers to better understanding the quality control or what the data is about. On the whole, this is a good manuscript that provides valuable data.

Response:

A Practical Applications subsection has been added to Line 219 discussing the use of ORACLES 4STAR data by Pistone et al. 2021, Cochrane et al. 2022, Fakoya et al. 2025, and a forthcoming publication by the coauthors.

RC3:

This manuscript presents a complete and consistent three-year dataset of 4STAR sky-scan measurements from the ORACLES 2016-2018 campaigns, addressing previous data gaps by resolving instrument artifacts and applying uniform quality control. To further enhance its quality and utility for researchers, consider expanding on the technical details behind the instrument issue resolution, particularly the scientific rationale for selecting the four specific wavelengths used for the 2017 and 2018 data, and briefly describing the applied calibration methods. Additionally, improve the clarity of the quality control (QC) criteria by providing more quantitative thresholds or logical rules, ideally complemented by illustrative examples showing the impact of QC on data, such as visualizing outlier removal. For improved data accessibility and usability, detail the specific file formats and variable naming conventions used in the archived datasets on ESPO and Zenodo. Finally, offer a quantitative summary of the dataset's

spatiotemporal coverage, including total measurement or flight hours and geographic ranges for each year, to help users quickly grasp the dataset's scale and characteristics.

Response:

The scientific rationale for the four-wavelength selection was added to Line 143. Discussion of calibration methods was added to Line 48. The QC criteria are further clarified by new Supplemental Tables 3-6. File formats have been added to Line 66, while the variable naming conventions are detailed in Supplemental Table 2. Spatiotemporal coverage is now detailed in Supplemental Table 1.

RC4:

The manuscript presents a valuable extension of the 4STAR retrievals to the ORACLES 2017 and 2018 campaigns by addressing instrument artifacts and establishing automated quality control standards. The work is well-motivated and provide important dataset for the study of biomass burning aerosols and their climate impacts. However, several areas require clarification to enhance the manuscript's scientific rigor and usability for potential data users.

1. The manuscript briefly mentions the shift from a five-wavelength to a four-wavelength set to avoid instrument artifacts but lacks a detailed theoretical discussion. The authors should clarify how the removal of 400 nm affects the inversion of aerosol's properties to help potential users better understand its quality. In addition, the authors should discuss the potential biases introduced by the 4wl set, particularly the noted slight decrease in SSA at 500 nm. Is this systematic?

2. The manuscript would benefit from a case study or example analysis showcasing how the dataset can be applied. The authors can consider include a brief case study of biomass burning event to illustrate its practical applications.

3. To enhance the dataset's accessibility, I recommend including a detailed summary table listing key metadata such as variables names, temporal and spatial coverage. This will allow users to quickly evaluate the data's applicability for their research needs.

Response:

1. A theoretical discussion regarding why 500 nm was the most affected wavelength, along with the systematic bias of the wavelength selection change was added to Line 173.

2. A Practical Applications subsection has been added to Line 219 discussing the use of ORACLES 4STAR data by Pistone et al. 2021, Cochrane et al. 2022, Fakoya et al. 2025, and a forthcoming publication by the coauthors.

3. Spatiotemporal coverage is now detailed in Supplemental Table 1, while variable naming conventions are detailed in Supplemental Table 2.

---

## Author Response (AR1)

Referee Comment #1:

The manuscript is well-written and clearly expresses the methodology and results. I appreciated the authors efforts to provide a comprehensive product for the entire 3 years of the campaign, especially given the issues with an instrument artifact. Providing the data both on the ESPO website and in Zenodo seemed like a logical choice to increase the accessibility of the data. Researchers familiar with the data would go to the ESPO website while others may find it in Zenodo. It would be good to provide more context about the data on the Zenodo page. I would recommend linking to this manuscript in the Zenodo metadata so that researchers can learn more about the context of the data.

Author's Response #1:

Thank you for highlighting ways that we can increase data accessibility. As such, we have now linked to the preprint DOI on Zenodo.

Manuscript Changes #1:

N/A

Referee Comment #2:

This manuscript provides the data from the NASA ORACLES airborne filed campaigns that deployed a 4STAR instrument onboard a P-3 aircarft to measure columnar optical properties of biomass burning aerosol smoke plumes over the Southeast Atlantic Ocean from 2016 to 2018. The authors well describe the wavelength selection and quality control criteria, however, I think it is better to show a period of the data or a single case for example may help the readers to better understanding the quality control or what the data is about. On the whole, this is a good manuscript that provides valuable data.

Author's Response #2:

Thank you for bringing attention to ways that we can increase audience understanding of the dataset. In alignment with both your comment and that of RC4, we are adding a subsection showcasing the use of our current and previous datasets within the scientific literature.

Manuscript Changes #2:

A Practical Applications subsection has been added to Line 238 discussing the use of the previous ORACLES 4STAR 2016 dataset by Pistone et al. 2019; along with the current ORACLES 4STAR 2016-2018 dataset by Jethva et al. 2024, Fakoya et al. 2025, and a forthcoming publication by the coauthors.

Referee Comment #3:

This manuscript presents a complete and consistent three-year dataset of 4STAR sky-scan measurements from the ORACLES 2016-2018 campaigns, addressing previous data gaps by resolving instrument artifacts and applying uniform quality control. To further enhance its quality and utility for researchers, consider expanding on the technical details behind the instrument issue resolution, particularly the scientific rationale for selecting the four specific wavelengths used for the 2017 and 2018 data, and briefly describing the applied calibration methods. Additionally, improve the clarity of the quality control (QC) criteria by providing more quantitative thresholds or logical rules, ideally complemented by illustrative examples showing the impact of QC on data, such as visualizing outlier removal. For improved data accessibility and usability, detail the specific file formats and variable naming conventions used in the archived datasets on ESPO and Zenodo. Finally, offer a quantitative summary of the dataset's spatiotemporal coverage, including total measurement or flight hours and geographic ranges for each year, to help users quickly grasp the dataset's scale and characteristics.

Author's Response #3:

Thank you for mentioning ways that we can increase dataset clarity. We had mentioned the scientific rationale for the five-wavelength selection but agree that reiterating it for the four-wavelength selection will further drive the message home. We also agree that the calibration methods should be mentioned. Performing additional sensitivity testing (such as outlier removal) is a little outside of the scope of the paper, but we hope that the application of the QC criteria is further clarified by our new supplemental tables. In alignment with both your comment and that of RC4, we also now discuss the file format, variable naming conventions, and the dataset's spatiotemporal coverage.

Manuscript Changes #3:

Discussion of calibration methods was added to Line 48:

"4STAR sky radiance measurements were calibrated in the laboratory using an NIST (National Institute of Standards and Technology) referenceable 12-lamp 36-inch integrating sphere (Brown et al., 2005). 4STAR direct beam measurements were calibrated via refined Langley regressions (Schmid and Wehrli, 1995) conducted at Mauna Loa Observatory, whereby ground-based solar irradiance measurements are normalized by top-of-atmosphere irradiance during clear-sky conditions, removing Rayleigh extinction and background atmospheric effects. Calibrations generally bracket the campaigns, with sky radiance calibrations in March 2016, November 2016, June 2017, and November 2017; and direct beam calibrations in June 2016, November 2016, May 2017, February 2018, and August 2018."

File format information has been added to Line 72:

"4STAR sky-scans are compiled into daily NetCDF-3 files and archived on the NASA Earth Science Project Office (ESPO) website. The newest versions of these datasets (presented in the Methods section) are also available on Zenodo (Mitchell, 2025), both in daily NetCDF-3 files, as well as individual NetCDF-3 files of each sky-scan."

The scientific rationale for the four-wavelength selection was added to Line 155:

"These four wavelengths were again selected to best align 4STAR with AERONET inversion wavelengths (440, 675, 870, 1020 nm), limited to the longest wavelength provided by the 4STAR UV/VIS (Ultraviolet/Visible) spectrometer of 995 nm, and avoiding the instrument artifact near 420nm by using 500 nm."

Spatiotemporal coverage is now detailed in Supplemental Table 1. Variable naming conventions are detailed in Supplemental Table 2. The QC criteria are further clarified by new Supplemental Tables 3-6, which show the QC status of every sky-scan.

Referee Comment #4.1:

The manuscript presents a valuable extension of the 4STAR retrievals to the ORACLES 2017 and 2018 campaigns by addressing instrument artifacts and establishing automated quality control standards. The work is well-motivated and provide important dataset for the study of biomass burning aerosols and their climate impacts. However, several areas require clarification to enhance the manuscript's scientific rigor and usability for potential data users.

1. The manuscript briefly mentions the shift from a five-wavelength to a four-wavelength set to avoid instrument artifacts but lacks a detailed theoretical discussion. The authors should clarify how the removal of 400 nm affects the inversion of aerosol's properties to help potential users better understand its quality. In addition, the authors should discuss the potential biases introduced by the 4wl set, particularly the noted slight decrease in SSA at 500 nm. Is this systematic?

Author's Response #4.1:

Thank you for emphasizing ways that we can better explain the effects of both the instrument artifact and our new four-wavelength selection. As such, we have added a discussion of both topics.

Manuscript Changes #4.1:

A theoretical discussion regarding why 500 nm was the most affected wavelength, along with the systematic bias of the wavelength selection change was added to Line 187:

"500 nm is most affected because it had previously been constrained by shorter wavelength values at 400 nm in the five-wavelength set but is now the shortest retrieved wavelength in the four-wavelength set. The small negative shift in SSA at 500 nm appears to be systematic, as the SSA difference for about 80 % of retrievals is less than zero. Thus, one should expect an average bias of about -0.004 when comparing SSA at 500 nm from the four-wavelength set to the five-wavelength set."

Referee Comment #4.2:

2. The manuscript would benefit from a case study or example analysis showcasing how the dataset can be applied. The authors can consider include a brief case study of biomass burning event to illustrate its practical applications.

Author's Response #4.2:

Thank you for highlighting ways that we can increase audience understanding of the dataset. In alignment with both your comment and that of RC2, we are adding a subsection showcasing the use of our current and previous datasets within the scientific literature.

Manuscript Changes #4.2:

A Practical Applications subsection has been added to Line 238 discussing the use of the previous ORACLES 4STAR 2016 dataset by Pistone et al. 2019; along with the current ORACLES 4STAR 2016-2018 dataset by Jethva et al. 2024, Fakoya et al. 2025, and a forthcoming publication by the coauthors.

Referee Comment #4.3:

3. To enhance the dataset's accessibility, I recommend including a detailed summary table listing key metadata such as variables names, temporal and spatial coverage. This will allow users to quickly evaluate the data's applicability for their research needs.

Author's Response #4.3:

Thank you for bringing attention to ways that we can increase data accessibility. In alignment with both your comment and that of RC3, we now showcase the variable naming conventions and the dataset's spatiotemporal coverage.

Manuscript Changes #4.3:

Spatiotemporal coverage is now detailed in Supplemental Table 1, while variable naming conventions are detailed in Supplemental Table 2.

Other Manuscript Changes:

We discovered that one research flight, 09 August 2017, was missing from our database. It has now been added, resulting in Version 4 on Zenodo. This also resulted in minor changes to Table 2, Figure 5, Supplemental Table 1, and Supplemental Table 5.